# Mixed methods study evaluating the implementation of the WHO hand hygiene strategy focusing on alcohol based handrub and training among health care workers in Faranah, Guinea

**Sophie Alice Müller**[ID][1]*, **Alpha Oumar Karim Diallo**[2], **Carlos Rocha**[1], **Rebekah Wood**[1], **Lena Landsmann**[3], **Bienvenu Salim Camara**[4], **Laszlo Schlindwein**[5], **Ousmane Tounkara**[2], **Mardjan Arvand**[3], **Mamadou Diallo**[2], **Matthias Borchert**[1]

1 Centre for International Health Protection, Robert Koch Institute, Berlin, Germany, 2 Faranah Regional Hospital, Faranah, Guinea, 3 Unit for Hospital Hygiene, Infection Prevention and Control, Robert Koch Institute, Berlin, Germany, 4 Centre National de Formation et de Recherche en Santé Rurale de Maferinyah, Maférinya, Guinea, 5 Global Health and Biosecurity, Robert Koch Institute, Berlin, Germany

* muellers@rki.de

**Data Availability Statement:** We would like to state, that we are not able to make our underlying

## Abstract

### Introduction

The most frequent adverse health events in healthcare worldwide are healthcare-associated infection. Despite ongoing implementation of the WHO multimodal Hand Hygiene (HH) Improvement Strategy, healthcare-associated infection rate continues to be twofold higher in low- than in high-income countries. This study focused on continued evaluation of HH compliance and knowledge. The mixed method approach, with inclusion of patients and care-givers, provided insight into challenges and facilitators of the WHO HH Improvement Strategy, and highlighted improvement points.

### Methods

An uncontrolled, before-and–after intervention, mixed methods study in Faranah Regional Hospital was conducted from December 2017 to August 2019. The intervention implemented the WHO HH Strategy including HH training for healthcare workers (HCWs), and the relaunch of the local production of alcohol-based handrub (ABHR). A baseline assessment of HH knowledge, perception and compliance of HCWs was done prior to the intervention and compared to two follow-up assessments. The second follow-up assessment was complemented by a qualitative component.

### Results

Overall compliance six months post-intervention was 45.1% and significantly higher than baseline but significantly lower than in first follow-up. Knowledge showed similar patterns of improvement and waning. The perception survey demonstrated high appreciation of the

data set publicly available for ethical and data protection reasons. The data contain potentially identifying information: our data have been collected from a small group of participants, and even data that are not directly identifying in combination become identifying (e.g. sex, profession, department ward, work description). This restriction to data availability has been imposed by RKI's Data Protection Office. Data requests may be sent to: Robert Koch Institute, Data Protection Office, c/o Ms Claudia Enge, Nordufer 20, D-13353 Berlin, datenschutz@rki.de.

**Funding:** This study was funded by the BMZ (Bundesministerium für Zusammenarbeit) as part of the GIZ (Gesellschaft für Internationale Zusammenarbeit) University and Hospital Partnerships in Africa (ESTHER) Program (Ensemble pour une Solidarité Thérapeutique Hospitalière en Réseau) (Award Number 81213469). The funders had no role in study design, data collection and analysis, decision to publish or preparation of the manuscript.

**Competing interests:** The authors have declared that no competing interests exist.

intervention, such as local production of ABHR. HCW's were concerned about overconsuming of ABHR, however simultaneous quantitative measurements showed that consumption in fact was 36% of the estimated amount needed for sufficient HH compliance. Potential fields for improvement identified by HCWs to enhance sustainability were permanent ABHR availability, having a dedicated person with ownership over continuous simulation HH trainings including simulations to improve technique.

## Conclusion

The study shows that the WHO multimodal HH strategy has a positive effect on HCW compliance and knowledge. Improvement points identified by local staff like sensitization on appropriate ABHR amount per HH action should be considered for sustainable HH improvement.

## Introduction

Patient safety is "the absence of preventable harm to a patient during the process of health care" [1] and a major challenge in public health. Healthcare-associated infections (HAI) are associated with a lack of awareness of hand hygiene (HH) standards, or the non-compliance with them [2]. Out of 100 hospitalized patients, ten will acquire one or more HAIs in low- and middle-income countries [3]. The WHO states that the HAI rate can be reduced by up to 55% through appropriate hand hygiene [4]. Therefore, the WHO launched the Multimodal Hand Hygiene Improvement Strategy in 2005. The strategy was increasingly implemented in Africa after the West African Ebola virus disease outbreak [5–7], often in settings facing challenges to hand hygiene, such as unsustainable supply of alcohol-based hand rub (ABHR), lack of running water and the difficulty of the application of "5 moments of hand hygiene" in overcrowded settings [8, 9]. Apart from the lack of appropriate infrastructure, other deterring factors previously identified are (i) belonging to a certain professional category (i.e. doctor, nursing assistant, physiotherapist, technician); (ii) working in specific care areas (i.e. intensive care, surgery, anesthesiology, emergency medicine); (iii) understaffing and over- crowding; and (iv) wearing gowns and/or gloves [10].

Despite ongoing implementations of the WHO multimodal HH Improvement Strategy, the HAI rate is still at least twofold higher in low- compared with high-income countries [8]. Further implementation research is thus necessary to explore possible barriers that affect the implementation or long-term effectiveness of the strategy.

Our study combines qualitative and quantitative research on the impact of the WHO Multimodal Hand Hygiene Improvement Strategy implemented in the Regional Hospital Faranah, Guinea.

This mixed method approach focused on continued evaluation of the strategy's long-term-effectiveness as well as the inclusion of patients and care-givers perspectives to illuminate factors necessary for sustainability of local production of ABHR and to maintain HCW HH knowledge and compliance. We aspired to gain an in-depth insight on HH, its challenges in a low-resource setting and sustainability improvement points.

## Materials and methods

### Study setting

The study was conducted as part of the PASQUALE (Partnership to Improve Patient Safety and Quality of Care) project, which responds to the first ("Clean Care is Safer Care") WHO

Global Patient Safety Challenge [11]. The project took place in the Faranah Regional Hospital (FRH), Guinea, a reference hospital for a population of 300,000 inhabitants employing 91 healthcare and administrative staff-members. When the project started, the hospital had insufficient access to ABHR and water, and no running water at all: water was fetched from a borehole and distributed to the various points of use by buckets. Our project made an effort to improve the water supply, which over time improved indeed, albeit independently from our effort.

While the project covered all components of the WHO strategy [12], this paper focusses on the evaluation of "System change–ABHR at point of care", and "Training and education".

## Quantitative study

The results of baseline and first follow-up of this study have been reported in detail elsewhere [13]. To show long-term trends, including potential waning of knowledge and compliance, some results of baseline and first follow-up are cited and compared to results from second follow-up in the discussion section. Previously published results of baseline and first follow-up are clearly referenced.

The project consisted of five phases: preparatory phase, baseline evaluation, intervention, first and second follow-up. Preparatory phase (Phase 0): In December 2017, a participatory needs assessment was conducted in the FRH with self-evaluation of the facility's hygiene level by the hospital director. Baseline evaluation (Phase I): From January to March 2018, a quantitative and qualitative HH baseline assessment took place. Intervention (Phase II): In December 2018, a tailored workshop on HH for healthcare workers (HCWs) was conducted and the local production of ABHR was reintroduced. Every HCW received a pocket bottle of 100 ml ABHR and every ward or consultation room a bottle of 500 ml ABHR. A bottle exchange station was installed at the local pharmacy. This paper focuses on the evaluation of the first two components of the multimodal WHO HH strategy: (1) system change, and (2) training and education [12]. But our intervention also addressed all other components: (3) monitoring and feedback, by direct HH observations and immediate feedback to the participants observed, (4) visual reminders in the workplace, by hanging up posters in the workplace and the distribution of flyers; and (5) creation of safety climate, by asking local staff for suggestion for improvement and patients for their personal opinion on HH. First follow-up (Phase III): From December 2018 to February 2019, the first quantitative follow-up assessment was done. The second quantitative follow-up assessment was complemented by a qualitative study (Phase IV) and was done six months after the training from June 2019 to July 2019.

For HH knowledge, a score was calculated equaling the number of correct answers (maximum score 25 points), this score was the primary outcome variable of the knowledge assessment. Wilcoxon rank-sum tests were performed to compare median knowledge score at first and second follow-up with the baseline and to compare the HH perception of HCWs at first and second follow-up with each other. HCWs' perception on the impact of project intervention was assessed in phase III and phase IV.

HH compliance was observed directly and overtly without prior announcement and was calculated as the number of HH actions performed divided by the number of opportunities requiring HH according to the WHO "5 Moments of HH". $\chi^2$ tests was applied to compare proportions such as, compliance at follow-ups to baseline and compliance at first to second follow-up. As most health care professionals had more than one HH opportunity, the observations were not independent. For confidentiality purposes workers were not identified during observation. To account for this lack of independence a design effect of two was assumed and accounted for by doubling the standard error [14], in the same way as it was done in a similar

study [15]. Two-tailed p-values less than 0.05 were considered statistically significant. Multivariable logistic regression was performed. The initial model included confounders proposed in the literature such as "type of ward", "hand hygiene indication" and "professional category". These confounders were maintained if the crude OR differed substantially from the adjusted one. All quantitative data was analyzed using Stata 15.2 (StataCorp LLc, College Station, Texas USA).

## Qualitative study

In phase IV, one of the authors [BSC], who was not previously known to participants, conducted a qualitative study with 22 in-depth semi-structured interviews (IDIs) and 6 focus group discussions (FGDs) with 4–8 participants each. In order to ensure maximum variation of the study population, potential study participants were purposely sampled by another author [AOKD] to separate between professions and to represent various services of the hospital with both genders and an array of patients and caregivers. Following a presentation of the qualitative research's objectives to all HCWs working at the hospital, participants of the IDIs were contacted individually before the interview to arrange interview appointments at their time and place of each participant's convenience. For the FGDs, an appointment was arranged at least one day in advance with each profession and/or service of the hospital in order to guarantee maximum participation. Both IDIs and FGDs were recorded and transcribed verbatim; they had a typical duration of 45 minutes and 60 minutes respectively. All field work, transcription and analysis were carried out in French; verbatim quotations were translated into English by the authors. Coding and thematic content analysis were performed using Microsoft Excel for Windows 10 following the thematic framework of the quantitative component. The analysis of the data concentrated on exploring perceptions and opinions about the project's impact on the knowledge, perception and compliance vis-à-vis HH. The thematic framework identification included perceptions on ABHR production and consumption, as well as prospects of sustainability. A preliminary analysis was done during the data collection phase, facilitating gradual adaptation of the questions and initial reach of data saturation.

## Ethics

Ethics approval was obtained from the *Comité National d'Ethique pour la Recherche en Santé*, Guinea (N˚: 016/CNERS/19). Every participant obtained information about the study and signed a consent form.

## Results

### Characteristics of participants

Table 1 provides a description of the participants. A total of 81 out of 84 (96.4%) participated in the second follow-up assessment with a proportion of 51.9% female participants. Healthcare workers were categorized into five professional groups, with "Other" consisting primarily of smaller professional groups, such as dentists, pharmacists, biologists, auxiliary nurses and students, were grouped together because of the small size of each professional group.

In total, 22 IDIs were carried out. Table 2 provides general information of the participants of the IDIs. Moreover, six FGDs were done, each with different professional groups (8 members of the operating theater, 8 nurses, 6 laboratory workers, 6 midwives, 5 doctors and 4 members of the maintenance team).

**Table 1. Study population of quantitative research.**

|  |  | 2nd Follow-up n (%) |
|---|---|---|
| Number of Respondents (N) |  | 81 |
| Gender: female |  | 41 (51.9) |
| Profession: |  |  |
|  | Medical doctor | 14 (17.2) |
|  | Nurse | 12 (15.2) |
|  | Midwife | 7 (8.9) |
|  | Technician | 5 (6.3) |
|  | Others | 41 (51.9) |
| Department: |  |  |
|  | Internal Medicine | 7 (8.9) |
|  | Surgery | 12 (15.1) |
|  | Emergencies | 7 (8.9) |
|  | Obstetrics | 12 (15.1) |
|  | Pediatrics | 10 (12.7) |
|  | Others | 31 (39.2) |

## Hand hygiene knowledge

**Quantitative.** The median knowledge score of the second follow-up was 16.0/25 (IQR 15.0–18.0). Compared to previously published assessment rounds, second follow-up was significantly lower than first follow-up (-3.0, p<0.001), but significantly higher than in baseline (+6.0, p<0.001, Table 3) [13].

All professional categories had significantly better knowledge in second follow-up compared to baseline, except the professional group "Other". A majority of HCWs reported to have been trained within the last three years (83.5% second follow-up). A sub-analysis of knowledge scores was conducted using only questionnaires from HCWs who had participated in baseline and both follow-ups. This sub-analysis of 22 HCWs showed the same tri-phasic trend with significant increase from baseline (median 14.0; IQR 12.0–15.0) to first follow-up

**Table 2. Study population of IDIs.**

|  |  | IDI HCW n (%) |  | IDI patients/caregivers n (%) |  |
|---|---|---|---|---|---|
| Number of Respondents (N) |  | 14 |  | 8 |  |
| Gender: female |  | 2 (14.3) |  | 5 (62.5) |  |
| Profession: |  |  |  |  |  |
|  | Medical doctor | 6 (42.9) | Public servant | 3 (37.5) |  |
|  | Nurse | 5 (35.7) | Seller | 2 (25.0) |  |
|  | Midwife | 1 (7.1) | Worker | 2 (25.0) |  |
|  | Technician | 1 (7.1) | Student | 1 (12.5) |  |
|  | Others | 1 (7.1) |  |  |  |
| Department: |  |  |  |  |  |
|  | Internal Medicine | 1 (7.1) |  | 1 (12.5) |  |
|  | Surgery | 4 (28.6) |  | 4 (50.0) |  |
|  | Emergencies | 1 (7.1) |  | N.A. |  |
|  | Obstetrics | 2 (14.2) |  | 2 (25.0) |  |
|  | Pediatrics | 1 (7.1) |  | 1 (12.5) |  |
|  | Others | 5 (35.7) |  | N.A. |  |

**Table 3. Median hand hygiene knowledge score (IQR), Regional Hospital Faranah; maximum score: 25.**

| | | 2nd Follow-up | Difference to baseline** | P-value*, comparison with baseline** |
|---|---|---|---|---|
| Overall Knowledge Score | | 16.0 (15.0–18.0) | + 6.0 | <0.001 |
| By professional categories | | | | |
| | Medical doctor | 17.0 (15.0–19.0) | + 7.5 | 0.001 |
| | Nurse | 16.0 (13.5–18.5) | + 7.5 | <0.001 |
| | Midwife | 16.0 (16.0–17.0) | + 4.0 | 0.017 |
| | Technician | 17.0 (16.0–18.0) | + 2.0 | 0.026 |
| | Other | 16.0 (14.0–18.0) | + 4.0 | 0.247 |

* p-value calculated with Wilcoxon rank-sum test

** baseline results previously published [13].

(median 20.5; IQR 19.0–22.0, p<0.001), significant increase from baseline to second follow-up (median 18.0; IQR 16.0–19.0, p<0.001) and significant decrease between the two follow-ups (p = 0.002). It was, therefore, decided to include all HCWs in the knowledge analysis (Table 3).

**Qualitative.** Participants reported that the PASQUALE project has improved the knowledge and practices of HH. HCWs said to have *"benefited from a theoretical and practical training"* (FGD 1, doctors, FRH, 13/08/2019) and to have learnt that *"ABHR is faster [than handwashing] and does not cause dryness"* (IDI 7, nurse, FRH, 13/08/2019). They also reported to have a new understanding that HH methods vary depending on the circumstances: *"thanks to PASQUALE I now know when to use ABHR, [and when to] wash my hands with soap or wear gloves"* (IDI 7, nurse, FRH, 13/08/2019).

## HCWs' and patients' perception of HH

**Quantitative.** A total of 79 perception questionnaires were collected in the second follow-up. 53.1% of HCWs rated reminders by patients to perform HH to be effective. Table 4 shows that a majority of participants in the second follow-up selected 7 on a Likert Scale of 1 to 7 indicating a fully affirmative answer on different aspects of the impact of the intervention. ABHR use and educational activities were rated to have had a positive impact throughout.

**Table 4. HCWs' perception about impact of intervention.**

| | 2nd Follow-up n (%)* |
|---|---|
| Number of Respondents (N) | 79 |
| Has the use of ABHR made hand hygiene easier to practice in your daily work? | 69 (88.5) |
| Is the use of ABHR well tolerated by your hands? | 67 (85.9) |
| Did knowing the results of hand hygiene observation in your ward help you to improve your hand hygiene practices? | 68 (87.2) |
| Has the fact of being observed made you paying more attention to your hand hygiene practices? | 63 (80.8) |
| Were the educational activities that you participated in important to improve your hand hygiene practices? | 69 (89.6) |
| Has the improvement of the safety climate (. . .) helped you personally to improve your hand hygiene practices? | 66 (84.6) |
| Has your awareness of your role in preventing HAIs by improving your hand hygiene practices increased during the current hand hygiene promotional campaign? | 63 (80.8) |

* results are shown as number of respondents out of total selecting seven on a seven-point Likert scale, indicating a fully affirmative answer.

**Qualitative.** In general, all participants gave a positive feedback on the intervention. HCWs felt that the language and content of the training were adequate and well-oriented to tackle the HH needs and challenges of the hospital. They considered the training to be very beneficial: *"the training sessions were very good because they even gave us questionnaires to reflect on our knowledge"* (IDI 10, surgeon, FRH, 13/08/2019). Participants agreed that producing ABHR locally improved its availability: *"when one speaks about a main strength [of the intervention], everything turns around the availability of the solution (. . .). This availability (. . .) provides quality of care"* (FGD 1, doctors FRH, 13/08/2019).

Participants also explored HH challenges encountered during the intervention. They identified as main challenge the insufficient access to running water in the entire hospital due to the intermittent failure of the borehole (FGD 1 and 2, nurses and doctors, FRH, 13/08/2019).

Patients stated that *"hygiene gives health"* and that respecting hygiene prevents them from contracting diseases (IDI 18, patient surgery, FRH, 11/08/2019). They reported that they were satisfied by the hygiene practices of HCWs but complained about the poor hygiene in the hospital environment, the abundance of insects inside wards and the unreliable electricity supply. Patient relatives stated that *"Patient care is very good, but there are too many mosquitoes."* (IDI 14, patient relative, FRH, 11/08/2019), and that *"there is no electricity and also there are too many mosquitoes"* (IDI 15, patient relative, FRH, 11/08/2019).

## Compliance

**Quantitative.** A total of 519 HH opportunities was assessed. HH compliance was 45.1% (95% CI 36.5–53.7) in second follow-up (S1 Table). Compared to previously published assessment round [13], compliance was significantly lower than in first follow-up (-26.4, $p<0.001$), but significantly higher than at baseline (+21.4, $p<0.001$). In the second follow-up compliance of "Technicians" was highest at 75.6% (95% CI 59.1–94.0) and compliance of "Midwifes" lowest at 13.3% (95% CI -22.3–48.9). Compliance was highest for the indication "after patient contact" (64.5%, 95% CI 48.7–80.3) and lowest for "before aseptic tasks" (25.0%, 95% CI -26.2–76.1). Compared to previously published baseline data, compliance of "Nurses", "Technicians" and "Others" was significantly higher than in baseline ($p = 0.002$, $P<0.001$, $p = 0.009$ respectively). A non-significant deterioration of 5.6 was seen in "Medical doctors" ($p = 0.439$). Compliance "after patient contact" and "after contact with patient surrounding" was significantly better that in baseline assessment ($p = 0.001$, $P<0.001$, respectively). No deterioration was observed for any indication.

In multivariable analysis, compliance was associated with the intervention showing an OR of 2.3 in the second follow-up compared to baseline (95% CI 1.2–4.6; $p<0.001$) after adjustment for profession and indication group.

**Qualitative.** The majority of HCWs interviewed during the 2nd follow-up identified HH as a routine practice that was reinforced by PASQUALE: *"the use of the alcohol-based solution, that is, hand friction has become a reflex for us"* (FGD 2, nurses, FRH, 13/08/2019). To describe the current situation, HCWs speak of a *"culture of hand hygiene"* (IDI 8, medical doctor, FRH, 18/08/2019). In that sense, HCWs explain that they have the motivation and discipline to do hand rubbing as indicated: *"Health workers are motivated; they have the awareness for hand hygiene (. . .)"* (IDI 8, medical doctor, FRH, 13/08/2019).

## ABHR production and consumption

**Quantitative.** The local pharmacy produces 10L batches of ABHR four times a month. The average monthly ABHR consumption in 2019 was at 26.9L for the entire hospital. The recommended minimum monthly consumption at FRH is 74L based on national guidelines [16],

assuming a disinfectant use of 3 ml per HH action, a minimum of eight HH actions per delivery, and a minimum of two HH actions per consultation. HCWs thus consumed only 36% of the recommended minimum ABHR quantity.

**Qualitative.** Participants of all FGDs concluded that the most important benefit of the project was the improved access to an easy-to-use and effective disinfectant. HCWs stated that they were *"proud"* of using a solution which was produced in their own facility and which fulfilled *"international standards"* (FGD 2, FRH, 13/08/2019). Furthermore, they report that *"the use [of ABHR] is so easy that everyone is used to it."* (FGD 2, nurses, FRH, 18/07/2019). In the FGDs, medical doctors described the ABHR solution as a *"weapon"* for infection prevention: *"But your weapon here is this little bottle that allows you to reduce the risks of contamination, but if you forget it, it's like a soldier who forgets his weapon in the middle of a war."* (FGD 1, medical doctors, FRH, 13/08/2019).

Nurses reported in the FGD, that the reintroduction of ABHR production was *"so successful"* that they had concerns about the *"excessive usage"*: *"they put a lot* [of ABHR] *in their hands, they even spread it in the palm of their hand as if it was water. That finishes very quickly the small 100 ml bottle"* (FGD 2, nurses, FRH, 18/07/2019). Following this rationale, participants feel the need to introduce a more economical usage policy (FGD 2, nurses, FRH, 13/08/2019).

In the FGD with doctors, concerns about the misuse of ABHR at home were mentioned. The private use was confirmed in individual interviews with a doctor (IDI 3, doctor, FRH, 13/08/2019) and a nurse: *"I'm protected* [against infections] *at the hospital and also at home because I pocket my small bottle. It is with me at home and here* [at the hospital]" (IDI 7, nurse, FRH, 13/08/2019).

As a hindering factor of ABHR use, participants generally agreed on the difficulty in its procurement during night and holiday shifts: *"we sometimes run out of ABHR; and that is because there are holidays. The supply point is closed and the pharmacist is absent"* (IDI 2, medical doctor, FRH, 13/08/2019). In terms of a sustainable behavior change, participants spoke of a disinfection culture: *"(. . .) and I would even say that it has become part of the culture to [clean hands] before and after the acts, or a contact [with a patient]"*(IDI 8, medical doctor, FRH, 18/07/2019).

## Improvement points and prospects of sustainability

HCWs suggested moving the ABHR bottle exchange facility from the pharmacy to the emergency department to overcome problems connected with opening hours. Due to a presumed excessive use of ABHR, nurses requested a further training on "rational" use of the solution: *"stop wasting ABHR (. . .) the use must be rational and indicated"* (FGD 2, nurses, FRH, 18/07/2019).

HCWs proposed to identify one hygiene focal point of the FRH whose duty would be to remind colleagues to obtain new ABHR flacons in time, and to organize HH trainings on a continuous basis. These trainings should address all HCWs, considering staff turnover and providing simulations related to WHO's "five moments of hand hygiene". Participants requested trainings on strategies for complying with HH rules during emergency situations (FGD 6, midwife, FRH, 18/07/2019). They found it necessary to increase knowledge transfer within teams by helping each other to improve HH compliance: *"it takes teamwork, for example when you see a colleague without his bottle, he must be reminded so it becomes for him a reflex before every act"* (FGD 1, medical doctors, 13/08/2019), *"the nurses are used to ABHR before an act, even when she [the nurse] does not have it [ABHR], she asks her colleague to put some solution on her hands"* (FGD 2, nurses, FRH, 18/07/2019). To sum up the improvement

points one participant said "*We must always make reminders and make friction solution available at all times.*" (IDI 9, medical doctor, 13/08/2019).

## Discussion

This study utilized a mixed method approach to assess the implementation of the WHO multi-modal HH Strategy in a low-resource setting. This assessment focus on the long-term trends and challenges to sustainability, complementing a previous study on short term changes [13].

The knowledge score reached 64%, thus showed a significant decrease compared to the previously published first follow-up. This decrease may be partially due to high staff turn-over, which is reflected by the decreasing proportion of previously trained HCWs. Nevertheless, knowledge at the second follow-up was still significantly higher than at baseline in all professional groups ("Medical Doctor" 13.5, "Nurse" 11.5, "Midwife" 13.0, "Technician" 13.0) except for "Others", consisting primarily of medical students and auxiliary nurses and presumably particularly affected by high staff turn-over [13]. The qualitative study also identified high staff turn-over as a problem. A HH study from Mali also showed a comparably low knowledge increase in nursing assistants, calling the attention to this professional group given its high number of HH opportunities [15]. A sub-analysis of knowledge scores was conducted using only questionnaires from HCWs who had participated in baseline and both follow-ups. This sub-analysis of 22 HCWs showed the same tri-phasic trend with significant increase from baseline to first follow-up, significant increase from baseline to second follow-up and significant decrease between the two follow-ups. It was, therefore, decided to include all HCWs in the full knowledge analysis. It can be theorized that similar results of the sub-analysis to the analysis of all participants are an effect of the culture of HH knowledge being shared between HCWs.

Perception of the WHO multimodal HH strategy was positive, especially on the component of local ABHR production and HH trainings for HCWs. The qualitative study confirmed these survey findings, and added evidence of high patient satisfaction with HH at FRH. The high appreciation of the project could partly be due to social desirability bias, a type of response bias previously reported in the Guinean context [17]. To minimize this bias the qualitative study was performed by a Guinean researcher [BSC], who was not project staff and not previously known to HCWs.

Compliance throughout the project timeline showed a threefold increase from baseline (23.7%) to first follow-up (71.5%) [13] and a significant decrease to 45.1% in the second follow-up six months post the intervention. To show trends, the second follow-up is compared to previously published assessment rounds. Fig 1 shows HH compliance overall and by indication.

HH compliance was higher "after patient contact" than "before patient contact", and remained significantly higher than at baseline only "after patient contact" and "after contact with patient surrounding", suggesting that HCWs may be primarily concerned about their own protection than about protecting patients [18]. An alarmingly pronounced drop "before aseptic tasks" was observed, calling for further attention in future trainings given the importance of this indication for patient safety.

Fig 2 shows overall compliance and compliance by profession. The overall compliance of the second follow-up is almost two times higher than in comparable settings seven months post intervention [5].

However, the significant decrease compared to first follow-up is a downwards trend that should be countered by considering suggestions for improvement identified by our qualitative research. Highest decrease in compliance was seen for the professional group of medical

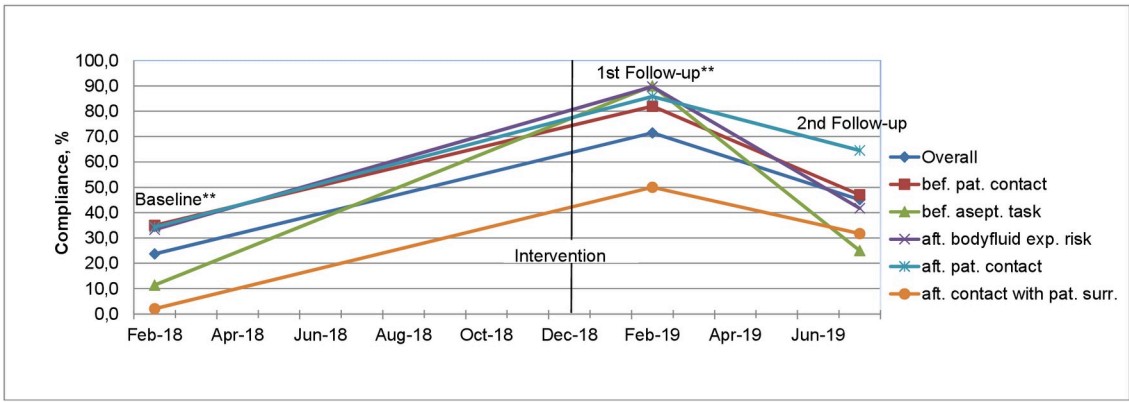

**Fig 1. Hand hygiene compliance at baseline, 1st and 2nd follow-up, overall and by indication.** ** Baseline and 1st follow-up data from previous publication [13].

doctors—a professional group known for its poor adherence following previous WHO studies [19]. The initial improvement for both compliance and knowledge may correspond to the first two phases of a tri-phasic learning pattern [20]. A tri-phasic learning curve is characterized by a rapid initial learning phase (right after the intervention), accompanied by a decline in the improvement and followed by a recovery to a steady state of improvement, which has already been described in a study assessing the learning curve of medical staff [20]. This tri-phasic pattern can be explained by the fact that new information can initially be quickly absorbed and reproduced, but its integration into everyday clinical practice occurs only gradually. Thus, the decline after initial improvement can be explained, which then leads to a steady state with increasing practice and the integration of new structures and methods. In our study both knowledge and compliance show signs of waning upon second follow-up. At least one further follow-up will be necessary to understand if the effect six months post intervention is the beginning of a steady state of improvement.

Taking into account the decrease in HH compliance and knowledge in the second follow-up, continuous and repeated trainings are thought to be beneficial to prevent further waning. Therefore, the hospital has introduced mini-trainings during staff meetings, as well as appreciation of good hygiene performance with the recognition of a "hygienist of the trimester". A

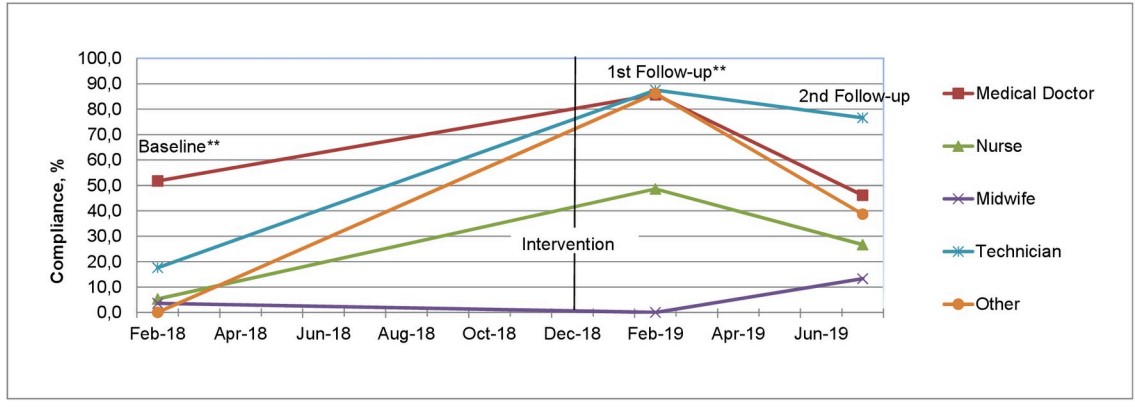

**Fig 2. Hand hygiene compliance at baseline, 1st and 2nd follow-up, by profession.** ** Baseline and 1st follow-up data from previous publication [13].

third follow-up will assess whether these measures can stabilize or even increase HH knowledge and compliance.

We used direct observations, as suggested by the WHO. However, its limited validity was addressed in a systematic review [21] where potential information and selection bias as well as confounding were discussed. The project team minimized information bias by thorough training of the local observer. Since direct observation was performed by a single researcher, inter-observer reliability was not an issue. The Hawthorne effect is believed to decrease over time as HCWs become more accustomed to being observed [22]. To minimize selection bias, HH compliance in the current study was not only assessed during normal working hours but also during night- and weekend-shifts. 86% of previous studies could not mitigate selection bias in this way [21]. Multivariable logistic regression was performed to control confounding. Profession group and indication were checked for multicollinearity to compliance and both variables were found to have a variance inflation factor (VIF) of 1.00, indicating no need for further investigation.

When addressing the same issues, qualitative and quantitative data came to the same conclusions, thus increasing validity by methodological triangulation. A further strength of the study is the high response rate of over 80% of HCWs in quantitative survey.

Qualitative analysis found a strong perceived need, in patients and HCWs alike, for infrastructural improvement to foster a safer hospital environment, requesting running water and renovation of wards. In the meantime, the hospital has been renovated by the Guinean Ministry of Health.

Our project emphasized the necessity to keep ABHR as easily accessible as possible to facilitate HH measures. The suggestion to move the ABHR supply facilities should be further explored, as it could also represent the search of an excuse for one's own lack of foresight and non-compliance. In high-income countries an ABHR provision alone could only marginally increase HH compliance, and behavioral modification programs were required [23]. In low resource settings, however, studies showed that the availability of ABHR is critical, and barriers to access ABHR flacons should be minimized [15, 24]. Interestingly, the established barriers for HH compliance of "extra workload" and "time constraints" were not mentioned by study participants [6], nor was the investment in patient safety seen as a way of achieving financial savings, and such savings were not considered a major contributor to HCW motivation. The qualitative research component of our study included patients' and caregivers' opinion on HH and patient safety. As HH is not only an issue for HCWs, viewpoints of patients and caregivers enriched our study. Nevertheless, HCWs in this study rate the impact of patient reminders as low in terms of HH improvement, and do not seem to be receptive to patients' and caregivers' input. The low literacy rate in Faranah may have led to this low appreciation of the patients' involvement.

Beyond the exploration of improvement points from the perspective of patients and HCWs, elucidating the HCW's misconception on the quantity of ABHR use is another example how qualitative research complemented the quantitative survey.

HCWs' take pride and ownership in the local production of disinfectant. This may have contributed to using it too sparingly in an effort to protect the supply. HCWs perceive ABHR as too precious to be used "*excessively*" and rate current consumption as too high. However, this perceived overconsumption does not match the average monthly consumption of 27L only, when 74L would be the estimated minimum consumption appropriate for the FRH. This low consumption rate can be due to either an inappropriate amount of ABHR used in each HH action, or missed HH actions. High HH compliance rates show that actions are being performed when needed, indicating that HCWs use less than the recommended amount of ABHR. HCW statements on concerns about economic and excessive usage further support

this theory. As such, future trainings should also focus on the proper amount of ABHR used in each HH technique. This misconception should be addressed in future training sessions at HRF; more research is needed to find out whether this phenomenon is prevalent in other settings as well, so that it can be addressed in the WHO HH strategies, if appropriate.

HCWs themselves asked for further training on HH including the correct amount and time for ABHR use. They suggested that this training should be a continuous activity conducted by a local HCW, showing the wish for local ownership and a motivation to maintain training efforts. HCWs identified improvement points to sustainably enhance prospects of high compliance, including simulations of daily hospital scenarios. This request corresponds to the 2019 WHO approach on "Scenario-based simulation training"[25].

## Conclusion

The current study suggests that especially the local production of ABHR and HH training of the WHO multimodal HH strategy may have a sustained positive effect on HCW compliance and knowledge in the FRH, but additional assessments are needed to evaluate the long-term effectiveness. This study identified improvement points via a mixed methods approach. To further enhance prospects of sustainability, we recommend focusing on the sensitization on appropriate ABHR consumption, the permanent ABHR availability and continuous trainings with local ownership. When aiming at sustainable improvement of HH in similar settings, the identified improvement points should be taken into consideration.

## Supporting information

**S1 Table. Hand hygiene compliance at baseline and follow-up, Faranah Regional Hospital, Guinea.** * width of CI adjusted for lack of independence by inflating standard error by a factor of 2. ** determined by $\chi^2$ test comparing with baseline with inflated standard error. ***determined by $\chi^2$ test comparing with 1st follow-up with inflated standard error. (XLSX)

## Acknowledgments

We would like to thank Pimrapat Gebert for her statistical advice and all study participants at the HRF for taking part in our project.

## Author Contributions

**Conceptualization:** Sophie Alice Müller, Bienvenu Salim Camara, Mardjan Arvand, Matthias Borchert.

**Data curation:** Sophie Alice Müller, Alpha Oumar Karim Diallo, Carlos Rocha, Rebekah Wood, Lena Landsmann, Bienvenu Salim Camara, Ousmane Tounkara.

**Formal analysis:** Sophie Alice Müller, Carlos Rocha.

**Funding acquisition:** Sophie Alice Müller, Laszlo Schlindwein, Mamadou Diallo, Matthias Borchert.

**Investigation:** Sophie Alice Müller, Alpha Oumar Karim Diallo, Carlos Rocha, Rebekah Wood, Lena Landsmann, Bienvenu Salim Camara, Laszlo Schlindwein, Ousmane Tounkara, Mardjan Arvand, Mamadou Diallo, Matthias Borchert.

**Methodology:** Sophie Alice Müller, Carlos Rocha, Rebekah Wood, Lena Landsmann, Bienvenu Salim Camara, Laszlo Schlindwein, Mardjan Arvand, Matthias Borchert.

**Project administration:** Sophie Alice Müller.

**Supervision:** Sophie Alice Müller, Alpha Oumar Karim Diallo, Carlos Rocha, Mardjan Arvand, Mamadou Diallo, Matthias Borchert.

**Validation:** Sophie Alice Müller, Carlos Rocha, Rebekah Wood, Bienvenu Salim Camara, Mardjan Arvand, Mamadou Diallo, Matthias Borchert.

**Visualization:** Sophie Alice Müller, Carlos Rocha, Rebekah Wood, Bienvenu Salim Camara.

**Writing – original draft:** Sophie Alice Müller, Carlos Rocha, Rebekah Wood, Lena Landsmann.

**Writing – review & editing:** Sophie Alice Müller, Alpha Oumar Karim Diallo, Carlos Rocha, Rebekah Wood, Lena Landsmann, Bienvenu Salim Camara, Laszlo Schlindwein, Ousmane Tounkara, Mardjan Arvand, Mamadou Diallo, Matthias Borchert.

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
