## [Decision Letter · Decision Letter 0]

8 Jun 2021

PONE-D-21-00332

Mixed methods study evaluating the implementation of the WHO Hand Hygiene Strategy

PLOS ONE

Dear Dr. %Sophie Mueller%

Thank you for submitting your manuscript to PLOS ONE. After careful consideration, we feel that it has merit but does not fully meet PLOS ONE’s publication criteria as it currently stands. Therefore, we invite you to submit a revised version of the manuscript that addresses the points raised during the review process.

We look forward to receiving your revised manuscript.

Kind regards,

Mary Hamer Hodges, MBBS MRCP DSc

Academic Editor

PLOS ONE

Additional Editor Comments (if provided):

As reviewer #2 has pointed out you have ovnly addressed one of five aspects of the WHO HH strategy. Please consider the others before resubmission.

Journal Requirements:

 “This study was funded by the BMZ (Bundesministerium für Zusammenarbeit) as part of the GIZ (Gesellschaft für Internationale Zusammenarbeit) University and Hospital Partnerships in Africa (ESTHER) Program (Ensemble pour une Solidarité Thérapeutique Hospitalière en Réseau) (Award Number 81213469). The funders had no role in study design, data collection and analysis, decision to publish or preparation of the manuscript.”

a) Please provide an amended Funding Statement that declares *all* the funding or sources of support received during this specific study (whether external or internal to your organization) as detailed online in our guide for authors at http://journals.plos.org/plosone/s/submit-now. 

b) Please state what role the funders took in the study.  If any authors received a salary from any of your funders, please state which authors and which funder. If the funders had no role, please state: "The funders had no role in study design, data collection and analysis, decision to publish, or preparation of the manuscript."

Please send your amended statements by return email; we will change the online submission form on your behalf.

**a)** If there are ethical or legal restrictions on sharing a de-identified data set, please explain them in detail (e.g., data contain potentially sensitive information, data are owned by a third-party organization, etc.) and who has imposed them (e.g., an ethics committee). Please also provide contact information for a data access committee, ethics committee, or other institutional body to which data requests may be sent.

**b)** If there are no restrictions, please upload the minimal anonymized data set necessary to replicate your study findings as either Supporting Information files or to a stable, public repository and provide us with the relevant URLs, DOIs, or accession numbers. For a list of acceptable repositories, please see http://journals.plos.org/plosone/s/data-availability#loc-recommended-repositories.

5. Please upload a copy of Supporting Information S1 Table which you refer to in your text on page 25.

Reviewers' comments:

Reviewer's Responses to Questions

**Comments to the Author**

1. Is the manuscript technically sound, and do the data support the conclusions?

Reviewer #1: Yes

Reviewer #2: Partly

2. Has the statistical analysis been performed appropriately and rigorously? 

Reviewer #1: Yes

Reviewer #2: Yes

3. Have the authors made all data underlying the findings in their manuscript fully available?

Reviewer #1: Yes

Reviewer #2: Yes

4. Is the manuscript presented in an intelligible fashion and written in standard English?

Reviewer #1: Yes

Reviewer #2: Yes

5. Review Comments to the Author

Reviewer #1: Full Title: Reading through, the study is among “health workers” i suggest you include that please. I suggest you including study location too as the full title and short title still have same word count.

Abstract

L12: I suggest including that baseline here

L18: I suggest you include the word “of”

Conclusion

L3: This abbreviation has not been defined

Introduction

L23: Again, this is not defined, if it is referring alcohol-based hand-rub, define it first and continue using the abbreviation throughout.

L24: I suggest you continue using HH since it has be defined or stay with household through the manuscript rather than mixing the abbreviations with the full word.

Study Settings

L7: I suggest including literature on other sources and access of water in the hospital if there is no water running system.

Quantitative study

L20: This definition and abbreviation should have come above, as the abbreviation is used in the abstract and L23 in the introduction already.

Qualitative study L11: How many participant per FGDs?

Hand Hygiene Knowledge

L13: Typo, the letter "e" is left out

Reviewer #2: To be clear, below are the 5 WHO Hand Hygiene (HH) Strategies

1a. System change – alcohol-based handrub at point of care

1b. System change – access to safe, continuous water supply, soap and towels

2. Evaluation and feedback

3. Training and education

4. Reminders in the workplace

5. Institutional safety climate Baseline evaluation Implementation Follow-up evaluation Review and planning

TITLE: The topic seem to have a different understand as what’s in the context. The study is talking about the whole HH strategy of WHO while the focus is mostly on the first strategy “alcohol based handrub at point of contact. Again, study location should have been included in the topic to at least make it a bit clear. In that case, I advised you look and revise the topic.

ABSTRACT

L18, 19 and 21 alcohol -based handrub was used without the abbreviation. However, in conclusion L3 the abbreviation (ABHR) was used. Please correct this, the abbreviation should be introduced in the first instance (L18) and only used thereafter. This process must be repeated in the main text so that you do not interchange alcohol-based handrub with ABHR.

INTRODUCTION

L3-4 “Healthcare-associated infections (HAI) are the most frequent adverse events worldwide.” This statement is very odd. Frequent adverse event following what? Hospital admission? Reference cited is old. From WHO fact sheet, there was specific case as to Healthcare Delivery Worldwide. So there should have been a focus on the refusal /omission of compliance or lack of knowledge of HH as a cause HAI.

L9 lack of running water in the facility, is that a reason for the research to focus mainly on only one of the 5 WHO HH strategies?

L21, 22, 23-24 show the research focus which is “to illuminate factors necessary for sustainability of local production of ABHR and to maintain HCW HH knowledge and compliance.” This clearly show that the key to the research is the first WHO HH strategy not the other four.

MATERIALS AND METHODS

Study Settings

L7 states that “the hospital lacks access to running water …” Is this the reason why Hand washing using soap, water and towel was not included, which is part 1b of the key WHO HH strategy?

Quantitative Study

L10-11 states that “The design, methods of the quantitative part and results of baseline and first follow-up of this study have been described in detail elsewhere.” Please indicate the where the details can be found? This is confusing!

L8-9 states that perception was compared with Wilcoxon rank-sum on phase III and IV while L10-13 stated that compliance was compared using WHO 5 Moment of HH strategy. Why are these different?

Qualitative study

L2 “In-depth semi-structured interviews and focus group discussions.” Both introduced without abbreviations but abbreviations were used in L6 and L8 respectively.

RESULTS

Hand Hygiene Knowledge

Qualitative

L8 states that one of the benefit form theoretical practical training was that “ABHR is faster (than handwashing) and does not cause dryness” Is this another reason for not considering hand washing as a key component of the strategy?

DISCUSSION

L7-8 states that “Nevertheless, knowledge at the second follow-up was still significantly higher than at baseline…” what is the baseline figure?

L19 states that “perception of the WHO multimodal HH strategy was positive throughout” This statement is not justified considering the fact that this study focuses mostly on just one aspect of the strategy

CONCLUSION

L20 states that “the WHO multimodal HH strategy may have a sustained positive effect on HCW compliance and knowledge in the FRH.” The positive effect suggested was mostly for 1a strategy: ABHR. Please review, perhaps describe the 5 point strategy first before focusing on only on point 1a.

6. PLOS authors have the option to publish the peer review history of their article (what does this mean?). If published, this will include your full peer review and any attached files.

Reviewer #1: No

Reviewer #2: No

---

## [Author Response · Author response to Decision Letter 0]

25 Jun 2021

Reviewer #1: Full Title: Reading through, the study is among “health workers” i suggest you include that please. I suggest you including study location too as the full title and short title still have same word count.

- Thank you, the new full title is “Mixed methods study evaluating the implementation of the WHO Hand Hygiene Strategy focusing on local production alcohol based handrub and training among health care workers in Faranah, Guinea” and short title “Mixed methods study evaluating the implementation of the WHO Hand Hygiene Strategy”

Abstract

L12: I suggest including that baseline here

- Thanks for this comment. However, baseline data have been published elsewhere, so we have removed them from this manuscript upon request by an earlier reviewer and the journal’s editor. Instead, we provide the reference in the manuscript where baseline results can be found: Müller et al. 2020 doi: 10.1186/s13756-020-00723-8

L18: I suggest you include the word “of”

- Thank you, we now state: “HCW’s were concerned about overconsuming of ABHR”

Conclusion

L3: This abbreviation has not been defined

- Thank you for spotting this mistake, abbreviation is now defined in L12 “(…) the relaunch of the local production of alcohol-based handrub (ABHR).”

Introduction

L23: Again, this is not defined, if it is referring alcohol-based hand-rub, define it first and continue using the abbreviation throughout.

- Thank you for spotting this mistake, abbreviation is now defined in L10 “(…) unsustainable supply of alcohol-based hand rub (ABHR),”

L24: I suggest you continue using HH since it has be defined or stay with household through the manuscript rather than mixing the abbreviations with the full word.

- Thank you, we now use the acronym HH for hand hygiene throughout the manuscript

Study Settings

L7: I suggest including literature on other sources and access of water in the hospital if there is no water running system. 

- We added more details on page 5 L7ff: “When the project started, the hospital had insufficient access to ABHR and water, and no running water at all: water was fetched from a borehole and distributed to the various points of use by buckets.”

Quantitative study

L20: This definition and abbreviation should have come above, as the abbreviation is used in the abstract and L23 in the introduction already.

- Thank you, we introduced the abbreviation in L10 in the introduction. 

Qualitative study L11: How many participant per FGDs?

- Thank you, we added “study with 22 in-depth semi-structured interviews (IDIs) and 6 focus group discussions (FGDs) with 4-8 participants each” in p7 L11ff and more detailed information can be found in p9, L 12ff “Moreover, six FGDs were done, each with different professional groups (8 members of the operating theater, 8 nurses, 6 laboratory workers, 6 midwives, 5 doctors and 4 members of the maintenance team).”

Hand Hygiene Knowledge

L13: Typo, the letter "e" is left out

- Thank you, we corrected this typo.

Reviewer #2: To be clear, below are the 5 WHO Hand Hygiene (HH) Strategies 1a. System change – alcohol-based handrub at point of care 1b. System change – access to safe, continuous water supply, soap and towels 2. Evaluation and feedback 3. Training and education 4. Reminders in the workplace 5. Institutional safety climate Baseline evaluation Implementation Follow-up evaluation Review and planning

TITLE: The topic seem to have a different understand as what’s in the context. The study is talking about the whole HH strategy of WHO while the focus is mostly on the first strategy “alcohol based handrub at point of contact. Again, study location should have been included in the topic to at least make it a bit clear. In that case, I advised you look and revise the topic.

- Thank you for this important point. You are right: in our paper we do not cover all components of the WHO strategy. Instead, we focus on (1) system change and (2) training - components where changes were most straightforward to measure. For your information: in the intervention: we also addressed (3) monitoring of practices and provision of feedback; (4) visual reminders in the workplace; and (5) creation of a safety climate within the institution. 

We, therefore, clarified in the title “Mixed methods study evaluating the implementation of the WHO Hand Hygiene Strategy focusing on local production alcohol based handrub and training among health care workers in Faranah, Guinea”. We added further details to the method section p6 L5ff: “This paper focuses on the evaluation of the first two components of the multimodal WHO HH strategy: (1) system change, and (2) training and education [12]. But our intervention also addressed all other components: (3) monitoring and feedback, by immediate HH observations and direct feedback to the participants observed, (4) visual reminders in the workplace, by hanging up posters in the workplace and the distribution of flyers; and (5) creation of safety climate, by asking local staff for suggestions for improvement and patients for their personal opinion on HH.”

ABSTRACT

L18, 19 and 21 alcohol -based handrub was used without the abbreviation. However, in conclusion L3 the abbreviation (ABHR) was used. Please correct this, the abbreviation should be introduced in the first instance (L18) and only used thereafter. This process must be repeated in the main text so that you do not interchange alcohol-based handrub with ABHR.

- Thank you for spotting this mistake, abbreviation is now defined in L12 “(…) the relaunch of the local production of alcohol-based handrub (ABHR).” And used thereafter. In the main text the abbreviation is introduced in L10 and used thereafter. 

INTRODUCTION

L3-4 “Healthcare-associated infections (HAI) are the most frequent adverse events worldwide.” This statement is very odd. Frequent adverse event following what? Hospital admission? Reference cited is old. From WHO fact sheet, there was specific case as to Healthcare Delivery Worldwide. So there should have been a focus on the refusal /omission of compliance or lack of knowledge of HH as a cause HAI.

- Thank you, we changed the statement and reference accordingly: “Healthcare-associated infections (HAI) are associated with a lack of awareness of hand hygiene (HH) standards, or the non-compliance with them [2].”. 

L9 lack of running water in the facility, is that a reason for the research to focus mainly on only one of the 5 WHO HH strategies?

- As now clarified in method section p6 L5ff, our project addressed all components of the WHO HH strategy. Within the component “system change”, both sub-components 1) running water and 2) ABHR were addressed. 

1)Running water: “Our project made an effort to improve the water supply, which over time improved indeed, albeit independently from our effort.” (p 5 L 9ff). 

2) ABHR: To address the challenge of very unreliable ABHR supply within the hospital and the stated wish of the local staff for internal production of ABHR, our intervention focused on the local production of ABHR. However, we are well aware, that all WHO components complement and depend on each other. ABHR, therefore should not be seen as a substitute for reliable running water supply, but as one of the important HH strategy components to sustainably improve HH. 

L21, 22, 23-24 show the research focus which is “to illuminate factors necessary for sustainability of local production of ABHR and to maintain HCW HH knowledge and compliance.” This clearly show that the key to the research is the first WHO HH strategy not the other four.

- Our paper focuses on the evaluation of two of the five WHO HH components, whereas all components were addressed in the intervention. For further detail please see method section p6 l5ff. 

MATERIALS AND METHODS

Study Settings

L7 states that “the hospital lacks access to running water …” Is this the reason why Hand washing using soap, water and towel was not included, which is part 1b of the key WHO HH strategy?

- Apologies for not being clear enough in the paper: hand washing using soap, water and towel was included in the intervention, but was not the focus of this paper. Please also see answer to comment above. 

Quantitative Study

L10-11 states that “The design, methods of the quantitative part and results of baseline and first follow-up of this study have been described in detail elsewhere.” Please indicate the where the details can be found? This is confusing!

- An in-text reference, pointing to the publication where these details can be found, directly follows the quoted statement. The reference is: Müller et al. 2020 doi: 10.1186/s13756-020-00723-8

L8-9 states that perception was compared with Wilcoxon rank-sum on phase III and IV while L10-13 stated that compliance was compared using Chi square WHO 5 Moment of HH strategy. Why are these different?

- Thanks for asking this question. We used the Wilcoxon rank sum test for the comparison of medians, the Chi2-test for the comparison of proportions. Compliance was summarised by proportions, knowledge and perception by medians, whereby perception was not compared between assessment phases. The section now reads: “Wilcoxon rank-sum tests were performed to compare median knowledge score at first and second follow-up with the baseline and to compare the HH perception of HCWs at first and second follow-up with each other. HCWs’ perception on the impact of project intervention was assessed in phase III and phase IV. HH compliance was observed directly and overtly without prior announcement and was calculated as the number of HH actions performed divided by the number of opportunities requiring HH according to the WHO “5 Moments of HH”. Chi2 tests was applied to compare proportions such as, compliance follow-ups to baseline and compliance at first to second follow-up.” P6 L17ff

Qualitative study

L2 “In-depth semi-structured interviews and focus group discussions.” Both introduced without abbreviations but abbreviations were used in L6 and L8 respectively.

- Thank you, we introduced the abbreviations in L12: “conducted a qualitative study with 22 in-depth semi-structured interviews (IDIs) and 6 focus group discussions (FGDs) with 4-8 participants” and took out the introduction of these abbreviations on page 9 L 10. 

RESULTS

Hand Hygiene Knowledge

Qualitative

L8 states that one of the benefit form theoretical practical training was that “ABHR is faster (than handwashing) and does not cause dryness” Is this another reason for not considering hand washing as a key component of the strategy?

- Hand washing was considered a key component of the strategy, and the project undertook efforts to improve water supply at the hospital. From the next quote the reader can take that hand washing had its place in the strategy: “thanks to PASQUALE I now know when to use ABHR, [and when to] wash my hands with soap or wear gloves”. Still, one respondent mentioned as something he/she had learned that “ABHR is faster [than handwashing] and does not cause dryness”. We made no change to this section.

DISCUSSION

L7-8 states that “Nevertheless, knowledge at the second follow-up was still significantly higher than at baseline…” what is the baseline figure?

- Thank you, we added “(…)higher than at baseline in all professional groups (“Medical Doctor” 13.5, “Nurse” 11.5, “Midwife” 13.0, “Technician” 13.0) except for “Others”, consisting primarily of medical students and auxiliary nurses and presumably particularly affected by high staff turn-over [12].”

L19 states that “perception of the WHO multimodal HH strategy was positive throughout” This statement is not justified considering the fact that this study focuses mostly on just one aspect of the strategy

- Thank you, we specified in the method section that other aspects of the WHO multimodal HH strategy, such as workplace reminders, monitoring and feedback were also addressed in the intervention. We changed the statement in the discussion section to: “Perception of the WHO multimodal HH strategy was positive, especially on the component of local ABHR production and HH trainings for HCWs” p 17, L 20ff. 

CONCLUSION

L20 states that “the WHO multimodal HH strategy may have a sustained positive effect on HCW compliance and knowledge in the FRH.” The positive effect suggested was mostly for 1a strategy: ABHR. Please review, perhaps describe the 5 point strategy first before focusing on only on point 1a.

- Thank you, we introduced the 5 components in the methods section and specified how every component was or was not addressed by the intervention, the evaluation and this paper. L 20 in the conclusion was clarified to: “The current study suggests that especially the local production of ABHR and HH training of the WHO multimodal HH strategy may have a sustained positive effect on HCW compliance and knowledge in the FRH”.

---

## [Decision Letter · Decision Letter 1]

16 Aug 2021

Mixed methods study evaluating the implementation of the WHO Hand Hygiene Strategy focusing on alcohol based handrub and training among health care workers in Faranah, Guinea

PONE-D-21-00332R1

Dear Dr. %Sophie Mueller%,

We’re pleased to inform you that your manuscript has been judged scientifically suitable for publication and will be formally accepted for publication once it meets all outstanding technical requirements.

Kind regards,

Mary Hamer Hodges, MBBS MRCP DSc

Academic Editor

PLOS ONE

Additional Editor Comments (optional):

Reviewers' comments:

Reviewer's Responses to Questions

**Comments to the Author**

1. If the authors have adequately addressed your comments raised in a previous round of review and you feel that this manuscript is now acceptable for publication, you may indicate that here to bypass the “Comments to the Author” section, enter your conflict of interest statement in the “Confidential to Editor” section, and submit your "Accept" recommendation.

Reviewer #1: All comments have been addressed

Reviewer #2: All comments have been addressed

2. Is the manuscript technically sound, and do the data support the conclusions?

Reviewer #1: Yes

Reviewer #2: Yes

3. Has the statistical analysis been performed appropriately and rigorously? 

Reviewer #1: Yes

Reviewer #2: Yes

4. Have the authors made all data underlying the findings in their manuscript fully available?

Reviewer #1: Yes

Reviewer #2: Yes

5. Is the manuscript presented in an intelligible fashion and written in standard English?

Reviewer #1: Yes

Reviewer #2: Yes

6. Review Comments to the Author

Reviewer #1: (No Response)

Reviewer #2: The research seem good now and all comment made have been appropriately addressed. I think it's good to go now from my side.

7. PLOS authors have the option to publish the peer review history of their article (what does this mean?). If published, this will include your full peer review and any attached files.

Reviewer #1: No

Reviewer #2: No

---

## [Editor Report · Acceptance letter]

18 Aug 2021

PONE-D-21-00332R1 

Mixed methods study evaluating the implementation of the WHO Hand Hygiene Strategy focusing on alcohol based handrub and training among health care workers in Faranah, Guinea 

Dear Dr. Müller:

I'm pleased to inform you that your manuscript has been deemed suitable for publication in PLOS ONE. Congratulations! Your manuscript is now with our production department. 

Kind regards, 

on behalf of

Dr. Mary Hamer Hodges 

Academic Editor

PLOS ONE